# Similarities in the General Chemical Composition of Colon Cancer Cells and Their Microvesicles Investigated by Spectroscopic Methods-Potential Clinical Relevance

**DOI:** 10.3390/ijms21051826

**Published:** 2020-03-06

**Authors:** Joanna Depciuch, Bartosz Klębowski, Małgorzata Stec, Rafał Szatanek, Kazimierz Węglarczyk, Monika Baj-Krzyworzeka, Magdalena Parlińska-Wojtan, Jarek Baran

**Affiliations:** 1Institute of Nuclear Physics Polish Academy of Sciences, PL-31-342 Krakow, Poland; joanna.depciuch@ifj.edu.pl (J.D.); bartosz.klebowski@gmail.com (B.K.); magdalena.parlinska@ifj.edu.pl (M.P.-W.); 2Department of Clinical Immunology, Institute of Pediatrics, Jagiellonian University Medical College, PL-30-663 Krakow, Poland; malgorzata.stec@uj.edu.pl (M.S.); rafal.szatanek@uj.edu.pl (R.S.); kazimierz.weglarczyk@uj.edu.pl (K.W.); mibaj@cyf-kr.edu.pl (M.B.-K.)

**Keywords:** colon cancer, extracellular vesicles, tumor-derived microvesicles, FTIR spectroscopy, FT-Raman spectroscopy, HCT116 cells, SW480 cells, SW620 cells, LoVo cells

## Abstract

Colon cancer constitutes 33% of all cancer cases in humans and the majority of patients with metastatic colon cancer still have poor prognosis. An important role in cancer development is the communication between cancer and normal cells. This may occur, among others, through extracellular vesicles (including microvesicles) (MVs), which are being released by both types of cells. MVs may regulate a diverse range of biological processes and are considered as useful cancer biomarkers. Herein, we show that similarity in the general chemical composition between colon cancer cells and their corresponding tumor-derived microvesicles (TMVs) does exist. These results have been confirmed by spectroscopic methods for four colon cancer cell lines: HCT116, LoVo, SW480, and SW620 differing in their aggressiveness/metastatic potential. Our results show that Raman and Fourier Transform InfraRed (FTIR) analysis of the cell lines and their corresponding TMVs did not differ significantly in the characterization of their chemical composition. However, hierarchical cluster analysis of the data obtained by both of the methods revealed that only Raman spectroscopy provides results that are in line with the molecular classification of colon cancer, thus having potential clinical relevance.

## 1. Introduction

Colon cancer is one of the most common cancers worldwide and in the last 15 years, the number of patients with this type of cancer has increased by 60% [1,2,3]. In cancer development, the communication between cancer and normal cells plays a crucial role [4]. Recent studies have claimed that cells can communicate with each other, among others, through the release of membrane-enclosed particles called extracellular vesicles (EVs) [5,6,7,8,9,10]. EVs consist mainly of smaller exosomes and larger microvesicles that can travel through the body fluids, thus conveying functional information to distant organs [11]. The release of EVs to the extracellular environment may occur either, upon the fusion of multivesicular bodies (MVB) with the plasma membrane (exosomes) or by membrane budding (microvesicles) [12]. As tumor-derived microvesicles (TMVs) are capable of transferring oncogenic proteins and nucleic acids that modulate the activity of recipient cells and play decisive roles in tumorigenesis, tumor growth, progression, drug resistance, and metastasis, there are expectations that they could be useful cancer biomarkers [13,14,15,16,17,18,19]. Moreover, chemically, TMVs consist mainly of proteins and lipids, therefore, it is very important to know which proteins and lipids make up the TMVs structure [20]. For protein analysis, usually the enzyme-linked immunosorbent assays (ELISA), western blot, or mass were performed, while for lipid composition evaluation, chromatography or mass spectroscopy were used [21,22]. These methods, however, are expensive and the data are limited to defining one fraction only (e.g., proteins, lipids, or nucleic acids).

In this study, by using Raman and FTIR spectroscopy, we determined the general chemical composition of four colon cancer cell lines: HCT116, LoVo, SW480, and SW620, which differ in their aggressiveness/metastatic potential [23,24]. In parallel, we analyzed TMVs released by the corresponding parental cancer cells. We hypothesize that the chemical composition of TMVs is similar to the parental cells that release them, allowing the TMVs to act as specific tumor messengers, thus preparing the distant organs for metastases development.

## 2. Results

### 2.1. Tumor-Derived Microvesicles (TMVs) Characterization

TMVs released by four colon cancer cell lines HCT116, LoVo, SW480, and SW620 were isolated from culture supernatants by differential centrifugation. Cell lines differed in malignancy potential, as HCT116 was described as fast growing, SW480-having low metastatic ability, SW620-high metastatic ability, and LoVo-undifferentiated [25]. The isolated TMVs were characterized by western blot analysis and, as shown in Figure 1, TMVs from all colon cancer cell lines contained exosome and MV-associated markers Alix, CD63, flotillin-1, and CD9 [26].

For the size distribution measurements, nanoparticle tracking analysis (NTA) was used. As shown in Figure 2, the data indicate that a vast majority of the isolated TMVs consisted of EVs in the size range of 120-400 nm, confirming their heterogeneous composition, which includes both exosomes and MVs. Taken together, the size distribution and composition of protein markers support the presence of MVs, according to the MISEV2018 guidelines [27].

### 2.2. General Chemical Composition of the Colon Cancer Cell Lines and the Corresponding TMVs

By using two complementary spectroscopic techniques, FTIR and FT-Raman, the general chemical composition of the four colon cancer cell lines and their corresponding TMVs was determined. In the obtained FTIR (Figure 3) and Raman spectra (Figure 4), peaks specific for nucleic acids, polysaccharides, proteins, lipids, water and other compounds making up the cellular and TMVs structures were identified and described in Table 1. The quantitative (value of maximum absorbance) and qualitative (shape of spectra, shift, or absence of maximum absorbance) differences between the respective cell lines and their TMVs have been detected.

Comparing the respective cell lines and their TMVs, we observed similarities in the shape of the spectra as well as in the values of maximum absorbance between the HCT116 (Figure 3a), LoVo (Figure 3b), and SW620 (Figure 3d) cells and their corresponding TMVs in the Infrared (IR) range from 600 cm^−1^ to 3000 cm^−1^. In the case of TMV_SW480_ and SW480 cells (Figure 3c), differences in the maximum absorbance values between 1500 cm^−1^ and 1700 cm^−1^ and between 2700 cm^−1^ and 3000 cm^−1^ were noticed, however, the shape of the TMVs and cells spectra were similar. Moreover, when we compared the FTIR spectra of the cell lines and TMVs collected from the respective cell lines, differences in the shape of the spectra and values of maximum absorbance in the FTIR range between 3000 cm^−1^ and 3800 cm^−1^ were visible. Furthermore, the absence of peaks in the spectra of cell lines compared to those of the corresponding TMVs were noticed. In the FTIR spectra of TMV_HCT116_ and TMV_SW480_, absence of peaks corresponding to C–O and CH_3_ asymmetric stretching from lipids were noticed in comparison with the respective cell lines. The identified peaks for all the obtained spectra are presented in detail in Table 1.

The acquired Raman spectra showed differences in the intensity values as well as in the shape of the spectra between the respective cell lines and TMVs released by them. In the spectra obtained for the HCT116 cells, the Raman intensities in the analyzed range were higher than for their TMVs (Figure 4a). Moreover, differences in the shape of the spectra, especially in the Raman shift between 1250 cm^−1^ and 1450 cm^−1^ (protein functional groups range) and between 3000 cm^−1^ and 3500 cm^−1^, were noticed when we compared the TMVs_HCT116_ and the HCT116 cell lines. When comparing LoVo (Figure 4b) and SW620 (Figure 4d) cells with their TMVs, the Raman intensities in the ranges between 800 cm^−1^–1400 cm^−1^ and 2700 cm^−1^–3000 cm^−1^ were higher for cells, while in the ranges of 1400 cm^−1^–1700 cm^−1^ and 3000 cm^−1^–3500 cm^−1^ were higher for TMVs. In the case of SW480 cells and their TMVs (Figure 4c), higher Raman intensities in all analyzed ranges were seen in the spectrum of TMV_SW480_. Between LoVo and SW480 cells and their corresponding TMVs, we observed differences in the shape of the spectra and peak shifts in the Raman range corresponding to protein vibrations (1400–1700 cm^−1^). Moreover, in the Raman spectra of SW480 cells and TMV_SW480_, differences in the shape of the spectra and peak shifts originating from DNA, phospholipids, and protein vibrations (800–1100 cm^−1^) were visible. Furthermore, in the Raman spectra of TMV_HCT116_ and TMV_SW620,_ absence of the peak corresponding to the C–H groups from cholesterol and cholesterol ester were noticed in comparison to their respective cell lines. The peaks originating from the stretching vibrations of the CH, NH, and OH groups were not observed in the Raman spectrum of the LoVo cell line in comparison with the Raman spectrum of the TMV_LoVo._ The identified peaks are described in detail in Table 1 for all the obtained spectra.

To obtain information about the differences in the structure between cell lines and the corresponding TMVs released from them, the second derivative of the FTIR spectra was calculated (Figure 5). However, we obtained the second derivative of the FTIR spectra not for the entire IR range measured, but only for the regions of nucleic acids and phospholipids (~900–1250 cm^−1^), proteins (~1250–1700 cm^−1^), and lipid (~2700–3000 cm^−1^) vibrations, respectively.

While comparing the respective cell lines and their TMVs, the most visible, however very small, structural changes occurred in the IR region corresponding to lipids. Changes in the protein structure were also noticed, when the second derivatives of the FTIR spectra of the TMVs_HCT116_, TMVs_SW480_, and TMVs_SW620_ were compared with those of their respective cell lines. Furthermore, structural changes in the IR region corresponding to DNA and phospholipid functional group vibrations were observed when we compared the second derivative of the FTIR spectra obtained for the SW480 cell line and its respective TMVs. Indeed, it is well known that in the IR region between 1700–1600 cm^−1^, there is a vibration of the C=O group, which participates in the formation of intramolecular and intermolecular hydrogen bonds. In this case, the stretching vibrations of the C=O bands in the IR spectra are very sensitive to the formation and changes in the hydrogen bonds between peptide groups [44,45]. Intramolecular hydrogen bonds in the C=O H–N group form the α-helix conformation in proteins, whereas the intermolecular hydrogen bonds of the C=O H–N group create β-form or β-sheet structures. For α-helix structures, the frequencies of the C=O vibrational bands are in the range between 1660–1650 cm^−1^; for structures of the β-form, the frequencies are in the range between 1640–1620 cm^−1^, while for β-sheet structures, it was 1670–1690 cm^−1^ [46]. To determine the detailed structural differences between HCT116, LoVo, SW480, and SW620 cell lines and their respective TMVs, we performed the analysis of the component bands, which was obtained from the amide I FTIR region (1600–1700 cm^−1^). This analysis enabled the determination of the protein secondary structure by quantifying the percentage of α-helix and β-sheet structures.

Figure 6 shows seven component bands in HCT116, LoVo, SW480 cells, and six in the SW620 cells. In the case of the TMVs released from the SW480 and SW620 cells, respectively, the number of component bands was the same as in the parental cells. However, in TMV_HCT116_, one component band less was detected than in the HCT116 cells. Moreover, when comparing LoVo cells and the TMVs released by these cells, the TMV_LoVo_ showed one component band less than the parental LoVo cells. Additionally, the component band areas corresponding to the α and β structures were different when comparing cells with their TMVs. The peak area values, percentage, and ratio of α and β secondary protein structures obtained for the analyzed cell lines and their TMVs are described in Table 2.

Similarity in the value of α and β secondary protein ratio structures in tumor cell lines and corresponding TMVs is presented. High similarities in the ratio of α-helix/β-harmonica in HCT116, LoVo, and SW480 cells and their corresponding TMVs have been detected. A similar trend was also observed for the SW620 cells and corresponding TMVs, however, it was less significant.

### 2.3. Determination of Similarities Between TMVs and Cell Lines

To determine the similarities in the general chemical composition between the analyzed cell lines and their TMVs, we performed hierarchical cluster analysis (HCA) (Figure 7).

Cluster analysis performed from the Raman (Figure 7a) and FTIR measurements (Figure 7b) showed two main groups. However, depending on the method, the components in these groups were different. It is very interesting that for both Raman and FTIR spectroscopy, the TMVs were similar to their respective cell lines. Moreover, HCA analysis obtained from Raman data showed very high similarities between the SW480 and SW620 cell lines and the TMVs collected from them (the lowest value of Euclidean distance), while HCA analysis obtained from the FTIR data showed that SW620 cells and TMVSW620 were more similar to LoVo cells and TMV_LoVo_ (Euclidean distance around 0.25). When we compared Euclidean distance in HCA obtained for Raman and FTIR data, the lower value of this parameter was in FTIR HCA analysis (Figure 7b), which means, that FTIR spectra of the measured samples were more similar to each other than the Raman spectra. Moreover, the value of the Euclidean distance (~10.5) for the LoVo cells and TMV_LoVo_ in HCA analysis for Raman suggest that the chemical composition of these two samples was the most different to the other six samples.

To solve whether it is sufficient to detect only the differences between the studied colon cancer cell lines and TMVs released by them, PCA was used to analyze the obtained Raman (Figure 8a) and FTIR (Figure 8b) spectra. The Raman PCA plot showed no separation of the SW480, TMV_SW480_, SW620, and TMV_SW620_. Consequently, it can be concluded that the SW480 and SW620 cells as well as their corresponding TMVs are very similar to each other with respect to their chemical composition. Moreover, PCA analysis of the Raman spectra showed that it will be very difficult to distinguish the HCT116 cells and the TMVs collected from this cell line. On the other hand, PCA analysis obtained from the FTIR spectra (Figure 8b) showed that it is not possible to distinguish the FTIR spectra of all the analyzed cell lines and their corresponding TMVs. However, PCA analysis obtained from the FTIR spectra showed that it is possible to detect chemical differences among the HCT116, LoVo, SW480, and SW620 cell lines as well as between the TMVs released by them using FTIR spectroscopy. The sensitivity of our results was between 85% and 99%, and the specificity was over 80% for more than 90% of cases. The respective sensitivities and specificities for the relevant PCA-LDA results are presented in Table 3.

## 3. Discussion

TMVs play a crucial role in the communication between the tumor and its microenvironment, promoting tumor progression, and metastasis formation. However, the exact mechanism involving TMVs in these processes is still unknown. In the current study, the general chemical composition of TMVs and their parental cells was analyzed by FTIR and Raman spectroscopy, which, to our best knowledge, has never been done before. Moreover, the size distribution of the isolated TMVs was measured by NTA, revealing their heterogeneous size range (120–400 nm), which showed that the samples included both exosomes and MVs. This heterogenic composition was further confirmed by western blot analysis and the expression of proteins characteristic for both exosomes and MVs. The FTIR and Raman spectra of the HCT116 (Figure 3 and Figure 4, violet spectra), LoVo (Figure 3 and Figure 4, blue spectra), SW480 cells (Figure 3 and Figure 4, red spectra), and SW620 (Figure 3 and Figure 4, black spectra) showed similarities between the general chemical composition of the cells and the corresponding TMVs. In addition, the α-helix to β-harmonica ratios (Table 2) confirmed even further similarities between the cancer cells and their TMVs. These results are of relevance as deconvolution of the amide I region in the FTIR spectra (Figure 6) allowed us to obtain information on the secondary structure of the whole protein. Consequently, our results showed that the TMVs released from individual cells have very similar components of proteins, which could be useful in the identification of their molecular subtypes. In addition, different numbers of components, in particular for SW620 cells and TMV_SW620_ and differences in the α-helix/β-harmonica ratio were observed. This may be related to the type of the tested protein in the solution, its concentration, solution pH, or solvent [47]. The solvent (water) and pH of the solution were the same for each sample tested. Therefore, the differences in the number of the obtained component bands or in the α-helix/β-harmonica ratio can be interpreted as changes in the composition of the protein fraction and its concentration.

Interestingly, in the HCA analysis (Figure 7), we found that the created similarity groups were different when generated from the Raman or FTIR spectra. Additionally, the PCA analysis obtained from the Raman and FTIR measurements (Figure 8) showed different possibilities for distinguishing the TMVs and their respective cell lines. The reason could be the difference in the physical principles of both techniques: IR requires the change of the dipole moment, whilst Raman relies on the change of bond polarity [48,49]. On the other hand, for an IR detectable transition, the molecule must undergo dipole moment change during vibration. So, when a molecule is symmetrical (e.g., O_2)_, we cannot observe any IR absorption lines since the molecule cannot change its dipole moment. It has been observed that molecules with a strong dipole moment are typically hard to polarize [50,51]. Additionally, FTIR spectroscopy is very sensitive to the presence of water and may induce a background in the obtained spectra, which in turn may have an impact on the reliability of measurements. As biological samples consist mainly of water, our results indicate that Raman spectroscopy may be a more adequate and thus more reliable spectroscopic method for biological samples. These differences underlying the principles of each method may consequently lead to changes in the intensity of the functional groups measured in the same sample, depending on the method used. Consequently, when we used Raman/FTIR spectra for the statistical analysis, we could observe differences between the results obtained for Raman and FTIR spectroscopy (Figure 7 and Figure 8). Moreover, FTIR spectroscopy is sensitive to hetero-nuclear functional group vibrations and polar bonds, especially OH stretching in water. Raman, on the other hand, is sensitive to homo-nuclear molecular bonds. For example, using Raman spectroscopy, C–C, C=C, and C≡C bonds can be distinguished. Therefore, molecules that cannot be detected by one method can be easily detected by the other. This is another reason why we noticed differences in the PCA and HCA analysis for Raman and FTIR spectra. This different physical background of the measurement principle had no significant influence on the fact that Raman and FTIR spectroscopy showed similarities in the chemical composition between cell lines and corresponding TMVs. In the case of HCT116 cells and their corresponding TMVs, we think that the number of functional groups, which can change polarization after laser irradiation (source of the electromagnetic wave in Raman spectrometer), is probably different for HCT116 cells and TMV_HCT116_, which could result in the observed distance between the cells and TMVs in the HCA analysis.

Moreover, in the obtained spectra, some vibration peaks were absent while the peaks corresponding to functional groups within the measured samples have been identified. Therefore, using these two complementary methods, a whole chemical structure of analyzed samples was obtained. Differences in the peak-shifts as well as the absence of some peaks were observed in Figure 3 and Figure 4 document the differences in the chemical composition between analyzed samples.

The two isogenic colorectal cancer SW480 and SW620 cell lines were derived from the same patient, the first originating from the primary tumor, whereas the latter was from a metastatic lesion to the lymph node, therefore similarities between these cells and their corresponding TMVs could be expected, as evidenced by several papers [52,53,54,55]. In this study, we observed a chemical composition similarity between the SW480 and SW620 cell lines and TMVs collected from them (Figure 7a and Figure 8a). In the study by Ji et al., the authors examined the protein profiles of exosomes isolated from both the primary (SW480) and metastatic (SW620) human colorectal tumor cells [52]. Based on their findings, the authors concluded that the exosomes obtained from the two cell lines had 628 proteins in common and that 313 and 168 proteins were unique for the SW480 and SW620 exosomes, respectively. The authors also showed a selective enrichment of key metastatic factors and signal transduction molecules in exosomes derived from the SW620 tumor cells as compared to the ones isolated from SW480 tumor cells. The data presented in this report corroborate those from Ji’s paper; however, our study focused on the whole population of TMVs (exosomes and MVs), and not just exosomes, which was the case in Ji’s group. Moreover, the FTIR and Raman data represent the identification of the overall functional groups present in the colorectal cancer cells (HTC116, LoVo, SW480, SW620) and their respective TMVs, which may generate different results than protein profiling only. The recent identification of four consensus molecular subtypes (CMS) of colon cancer with distinguishing features has provided evidence that the expression subtypes have clinical relevance independent of cancer stage [53,56]. According to this classification, Berg et al., in a very elegant study, analyzed 34 colorectal cancer cell lines by multi-level data integration including targeted deep sequencing, DNA copy numbers, gene expression, microRNA (miRNA) expression, and protein expression [57]. In their study, eight cell lines were classified as CMS1-“immune”, nine as CMS2-“canonical”, six as CMS3-“metabolic”, and 10 as CMS4-“mesenchymal”. One cell line, derived from a neuroendocrine tumor, had a distinct gene expression profile [58]. They also used the PCA to classify the cell lines, taking into account the PC1 from mRNA expression data to single-sample gene set enrichment analysis (ssGSEA). Using this approach, they classified the cell lines with low PC1/high ssGSEA score as colon-like (mainly CMS2 and CMS3 subtype, with higher levels of gastro-intestinal marker genes) and undifferentiated (mainly CMS1 and CMS4 subtype, having higher epithelial-to-mesenchymal transition (EMT) signature score and increased expression of TGFβ induced genes). This was further confirmed by differential mRNA, miRNA, and protein expression analysis. All of the cell lines used in our study were classified in Berg’s report as undifferentiated, and three out of four (HCT116, SW480, and SW620) were classified as the CMS4 subtype. Only Lovo cells belonged to the CMS1 group. This molecular classification was fully reflected by the Raman spectroscopic analysis of cancer cells and their TMVs as performed in our study (Figure 7).

On the other hand, Pablo et al. used Raman spectroscopy to distinguish different colon cancer cell types including SW480 and SW620. They showed that these two cell lines could be distinguished by ratio of α:β secondary protein structure. Furthermore, the PCA-LDA analysis showed that by using this approach, it is possible to distinguish different types of colon cancer cell lines with an accuracy of 92.4 ± 0.4% [59]. These results correspond with the results obtained in our work. Additionally, in our study, a significant difference in the α:β ratio between the SW480 and SW620 cells was observed (Table 2). In addition, the PCA-LDA analysis showed very high probability for the distinction of different colon cancer types (Table 3).

To summarize, in this report, through the use of FTIR and Raman spectroscopy, we showed that the chemical composition of TMVs is similar to the composition of the cells from which they are released. We suggest that Raman as well as FTIR spectroscopies could find a potential application for the first rapid chemical identification of circulating TMVs, reflecting the composition of their parental cancer cells. It seems however, that only the data obtained from the Raman analysis were in line with the molecular classification of colon cancer, having thus a potential clinical relevance. Further studies involving other tumor cell lines and their respective TMVs using Raman and FTIR spectroscopies would provide additional information on the usefulness of these methods, which could potentially translate into the clinical setting (i.e., tumor metastasis, formation of metastatic niche).

## 4. Materials and Methods

### 4.1. Cell Culture

Colon cancer cell lines (LoVo, SW480, and SW620) were obtained courtesy of Prof. Caroline Dive, Paterson Institute for Cancer Research, University of Manchester. Human colon carcinoma cell line HCT116 was obtained from the American Type Culture Collection (ATCC, Manassas, VA, USA) and maintained according to the ATCC’s instructions. Briefly, HCT116 cells were cultured in McCoy’s 5A medium (Gibco, Paisley, UK), while LoVo, SW480, and SW620 cells were cultured in Dulbecco’s Modified Eagle Medium (DMEM) with high glucose content (Corning^TM^, Corning, NY, USA) in a 37 °C humidified atmosphere with 5% CO_2_. All media were supplemented with 10% fetal bovine serum (FBS, Biowest, Nuaille, France) and gentamicin (50 µg/mL), (PAN-Biotech, Aidenbach, Germany). The cells were cultured by bi-weekly passages and regularly tested for *Mycoplasma* sp. contamination with the PCR-ELISA kit (Roche, Mannheim, Germany), according to the manufacturers’ instructions.

### 4.2. Isolation of TMVs

Supernatants from well-grown cell cultures were collected, centrifuged at 2000× *g* for 20 min to remove cell debris and then centrifuged again at 50,000× *g* (RC28S, Sorvall, Newton, CT, USA) for 1 h at 4 °C. Pellets were washed twice in PBS to remove FBS and finally re-suspended in serum-free medium. Quantification of TMVs proteins was evaluated by the Bradford method (BioRad, Hercules, CA, USA). TMVs were tested for endotoxin contamination by the Limulus test, according to the manufacturer’s instruction (Charles River Laboratories, Inc., Wilmington, MA, USA) and stored at −20 °C until use. To simplify, TMVs were named according to their cell line of origin (e.g., TMVs released from HCT116 as TMV_HCT116_, from LoVo–TMV_LoVo_, from SW480–TMV_SW480_, and from SW620 as TMV_SW620_.

### 4.3. Western Blotting

To assess the presence of different EVs markers in TMVs, the western blotting technique was employed. TMVs were suspended in M-PER lysing buffer (Pierce, Rockford, IL, USA) containing the protease inhibitor cocktail (Roche). The concentration of samples was measured using the Bradford kit (Bio-Rad) as per the manufacturer’s instructions. A total of 20 μg of suspended TMVs was mixed with NuPAGE LDS Sample Buffer (4×) and NuPAGE Sample Reducing Agent (10×) (both Life Technologies, Carlsbad, CA, USA). Samples were heated (70 °C, 10 min) and electrophoresed in 12% polyacrylamide gel containing sodium dodecyl sulfate (SDS). Next, electrophoresed samples were transferred onto the polyvinylidene fluoride membrane (PVDF, Bio-Rad). Then, after blocking for 1 h at room temperature in Tris buffered saline (TBS) with 0.1% Tween-20 (Sigma, St. Louis, MO, USA) and 1% bovine serum albumin (BSA, Sigma), the membranes were incubated overnight at 4 °C with mouse mAb anti-Alix (3A9), rabbit mAb anti-CD9 (D801A), and anti-Flotillin-1 (D2V7J) (all from Cell Signaling, Beverly, MA, USA) diluted 1:1000 as well as rabbit polyclonal Ab anti-CD63 (SIGMA) diluted 1:2000. As a loading control, rabbit anti-GAPDH antibodies (Cell Signaling) diluted 1:5000 were used. After incubation, the membranes were washed in TBS supplemented with BSA and Tween-20 and incubated for 1 h in room temperature with appropriate secondary antibody: goat anti-mouse and goat anti-rabbit (al secondary antibodies were used in dilution 1:2500) conjugated with horseradish peroxidase (Santa Cruz Biotechnology). The protein bands were visualized with the SuperSignal West Pico Chemiluminescence Substrate kit (Pierce), according to the manufacturer’s protocol and analyzed with ChemiDoc system (Bio-Rad).

### 4.4. Nanoparticle Tracking Analysis (NTA)

Average, modal size, and size distribution of the TMVs were obtained using the NANOSIGHT LM10-HS488FT14 Nanoparticle Characterization System (Malvern Instruments, Malvern, UK). Briefly, 1 μL of the TMV suspension was diluted 1000× in filtered (0.22 μm) PBS to obtain the total sample volume of 1 mL. Next, approximately 700 μL of the sample was loaded manually into the measuring chamber using an insulin-type syringe, after which the syringe was mounted onto the pump and the sample was delivered at a constant flow rate of 80 units. Next, three one-minute videos were recorded by the sCMOS camera for each sample and used later for analysis using the NanoSight NTA 3.0 analytical software (Malvern Instruments).

### 4.5. Cell and TMVs Preparation for Fourier Transrorm IntraRed (FTIR) and-Raman Measurements

For FTIR and FT-Raman spectrum acquisition, cells in the concentration 10^8^ cell/mL were centrifuged for 5 min at 3000 rpm. Subsequently, the cells were washed three times in isotonic solution (NaCl, 0.9%) to ensure complete removal of trypsin and culture medium. Next, dense cell suspensions were placed onto an attenuated total reflection (ATR) crystal (FTIR spectroscopy) or CaF_2_ slides (Raman spectroscopy).

### 4.6. FTIR Spectroscopy

All measurements were carried out on an EXCALIBUR FTS-3000 spectrometer (Bio-Rad, Digilab, UK) at room temperature. All spectra were recorded by attenuated total reflection (ATR) with a ZnSn crystal. A total of 0.5 mL of cell containing solution was deposited on the ATR ZnSn crystal. FTIR spectra were recorded between 4000 and 500 cm^−1^. Each spectrum was obtained by averaging 64 scans recorded at a resolution of 4 cm^−1^. For statistical validation, three samples for each of the cell lines were measured. Moreover, each sample was measured three times. Baseline correction and vector normalization of the obtained spectra were performed using OPUS 7.0 software (Bruker Optik GmbH, Ettlingen, Germany).

### 4.7. FT-Raman Spectroscopy

FT-Raman spectra were recorded using a Nicolet NXR 9650 FT-Raman spectrometer (Thermo Fisher Scientific, USA) equipped with an Nd:YAG laser (1064 nm) and a germanium detector. Measurements were performed in the range from 150 to 3.700 cm^−1^ with a laser power of 1 W. Unfocused laser beam was used with a diameter of approximately 100 μm and a spectral resolution of 8 cm^−1^. Raman spectra were processed by the Omnic/Thermo Scientific software based on 128 scans. The obtained spectra were normalized using vector normalization in OPUS 7.0 software (Bruker Optik GmbH, Ettlingen, Germany). Moreover, each spectrum was smoothed using the Savitzky–Golay algorithm. The number of smoothing points was nine.

### 4.8. Deconvolution of Amide I Region (1600–1700 cm^−1^)

The secondary protein structure was analyzed by means of curve fitting using MagicPlot software (Magicplot Systems, Saint Petersburg, Russia). First, the second derivative spectra were calculated based on the ATR-FTIR spectra to determine the initial peak positioning for the curve fitting. The peaks were fitted using a Gaussian function and the area under the curve was considered at 100%. Each component was expressed as its percentage after fitting.

### 4.9. Data Analysis

For all obtained spectra, vector normalization and baseline correction were applied. These operations were performed using OPUS 7.0 software. Moreover, in each FTIR as well as Raman spectrum, vibrations corresponding to nucleic acid, phospholipids, proteins, and lipids were analyzed. The number of obtained data from FTIR and Raman spectra was very large, therefore, to determine the similarity between the cancer and control cells, principal component analysis-linear discriminate analysis (PCA-LDA) was performed. First, PCA was undertaken, which reduced the dimensionality and the number of data variables, while maintaining as much variance as possible. Next, with the use of the LDA method, a suitable model for grouping data based on independent variables (wavelengths) was developed. Then, the PCA-LDA procedure was performed based on the selected spectral regions: between 1250 cm^−1^–1700 cm^−1^ and between 2700 cm^−1^–3000 cm^−1^. PCA-LDA results had been verified by the leave-one-sample out cross-validated method, which involves leaving all spectra from a single sample out of the model once before assessing performance. This analysis was done using the OriginPro 2015 software (OriginLab Co.; Northampton, MA, USA). Moreover, to determine similarities between HCT116, LoVo, SW480, SW620 cell lines, hierarchical cluster analysis (HCA) with Euclidean distance was performed in Past software (developed by Oyvind Hammer). HCA analysis was done for the fingerprint region (between 800 cm^−1^ and 1800 cm^−1^) for both the Raman and FTIR spectra of the cells and TMVs. For this purpose, 518 points from the FTIR spectra and 259 points from the Raman spectra were analyzed using the paired group (UPGMA) algorithm.

## Figures and Tables

**Figure 1 ijms-21-01826-f001:**
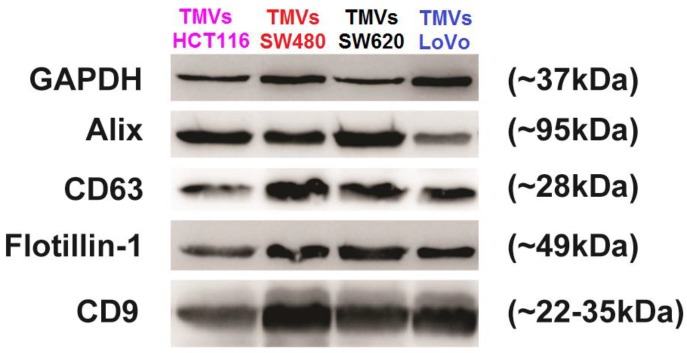
Western blot analysis of GAPDH, Alix, CD63, Flotillin-1, CD9 expression in TMVs collected from HCT116, LoVo, SW480, and SW620 cell lines. A total of 20 μg of the TMV protein was used for each run. After SDS-PAGE separation and transfer of the proteins onto the polyvinylidene fluoride membranes, the membranes were stained with mouse anti-Alix (3A9), rabbit anti-CD9 (D801A), anti-Flotillin-1 (D2V7J) mAbs, and rabbit polyclonal anti-CD63 Abs. As a loading control, rabbit anti-GAPDH Abs were used. Appropriate secondary Abs: goat anti-mouse and goat anti-rabbit conjugated with horseradish peroxidase, followed by SuperSignal West Pico Chemiluminescence Substrate Kit were used to visualize corresponding protein bands.

**Figure 2 ijms-21-01826-f002:**
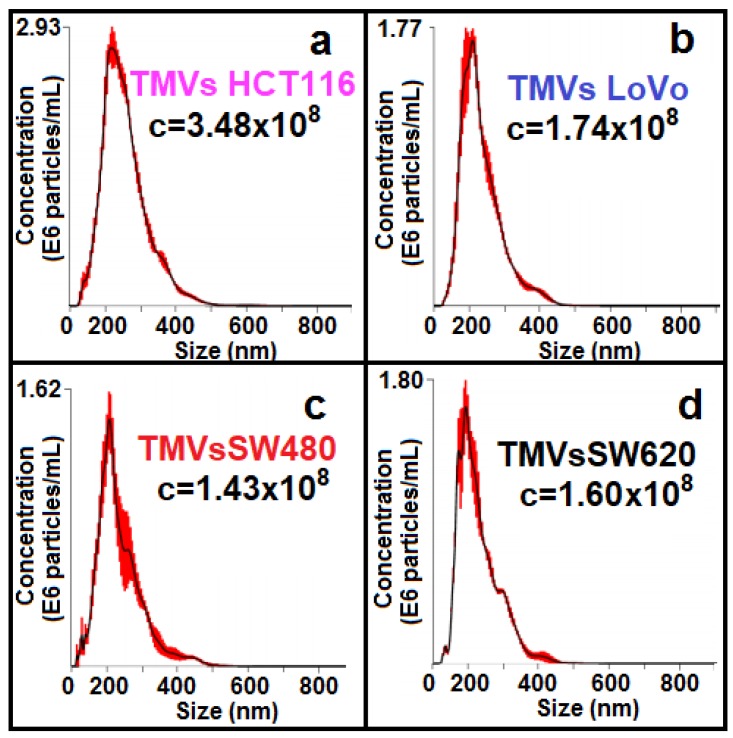
Nanoparticle tracking analysis of size distribution and concentrations (number EVs/mL) of TMVs collected from HCT116 (**a**), LoVo (**b**), SW480 (**c**), and SW620 (**d**) cell lines.

**Figure 3 ijms-21-01826-f003:**
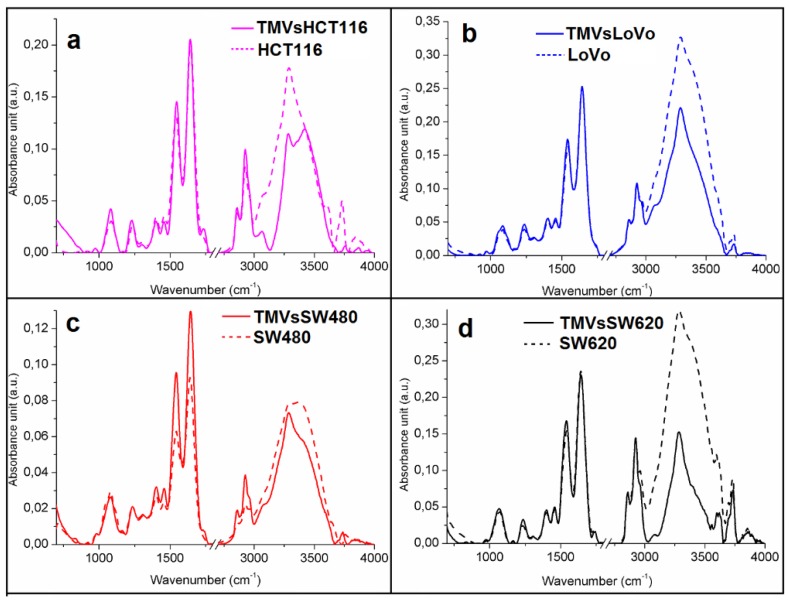
Offset FTIR spectra of: (**a**) HCT116; (**b**) LoVo; (**c**) SW480 and (**d**) SW620 cell lines (dashed lines) and of: (**a**) TMV_HCT116_; (**b**) TMV_LoVo_; (**c**) TMV_SW480_ and (**d**) TMV_SW620_ (full lines).

**Figure 4 ijms-21-01826-f004:**
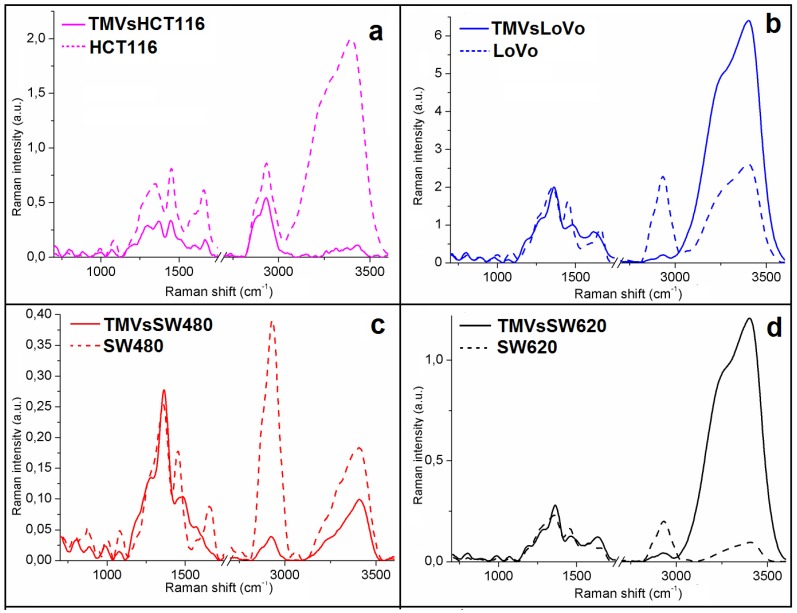
Raman spectra of: (**a**) HCT116; (**b**) LoVo; (**c**) SW480 and (**d**) SW620 cell lines (dashed lines) and of: (**a**) TMV_HCT116_; (**b**) TMV_LoVo_; (**c**) TMV_SW480_ and (**d**) TMV_SW620_ (full lines).

**Figure 5 ijms-21-01826-f005:**
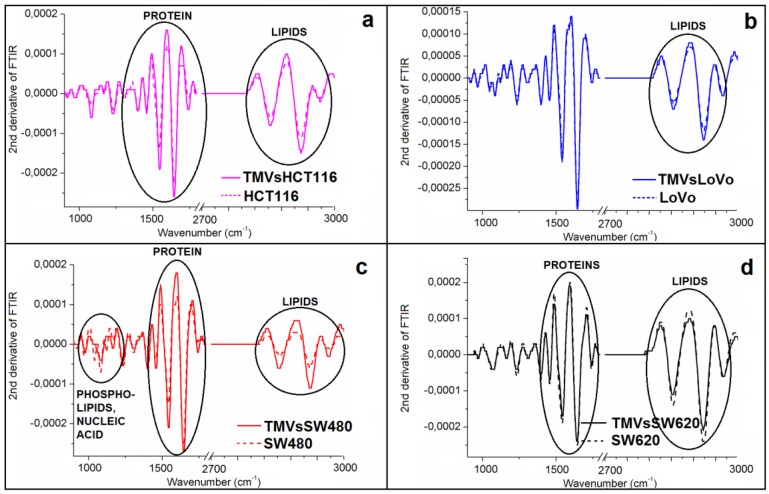
Second derivative of the FTIR spectra of: (**a**) HCT116; (**b**) LoVo; (**c**) SW480 and (**d**) SW620 cell lines (dashed lines) and of: (**a**) TMV_HCT116_; (**b**) TMV_LoVo_; (**c**) TMV_SW480_ and (**d**) TMV_SW620_ (full lines).

**Figure 6 ijms-21-01826-f006:**
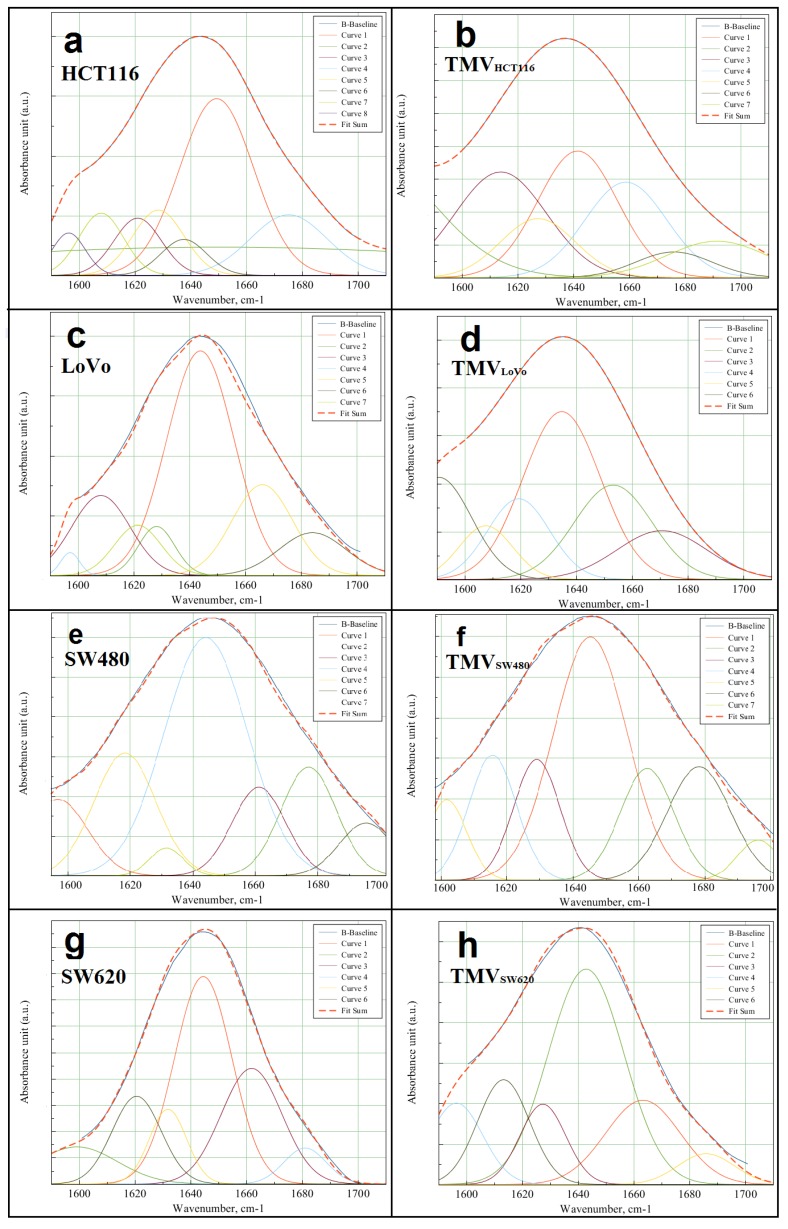
Deconvolution of amide I FTIR region (1700–600 cm^−1^) obtained for cell lines: (**a**) HCT116; (**b**) TMV_HCT116_; (**c**) LoVo; (**d**) TMV_LoVo_; (**e**) SW480; (**f**) TMV_SW480_; (**g**) SW620 and (**h**) TMV_SW620_. Curves 1–8 show the matched Gaussian functions and fit sum-results obtained by the matched Gaussian functions.

**Figure 7 ijms-21-01826-f007:**
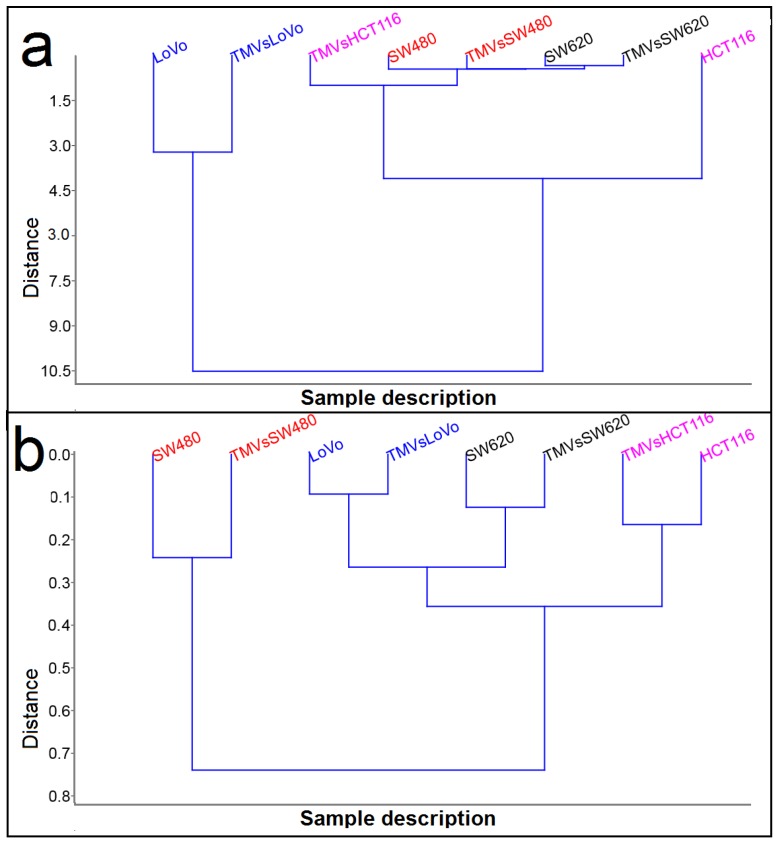
Hierarchical cluster analysis (HCA) of HCT116, LoVo, SW480, and SW620 cell lines and their corresponding, released TMVs from spectroscopy data: Raman (**a**) and FTIR (**b**).

**Figure 8 ijms-21-01826-f008:**
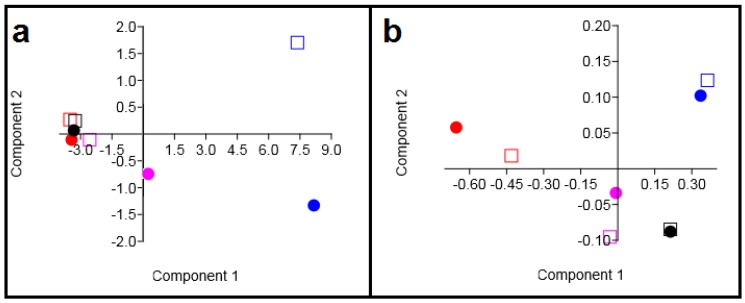
Principle component analysis of HCT116 (violet), LoVo (blue), SW480 (red), and SW620 (black) cell lines (dot) and their corresponding, released TMVs (square). Two-dimensional (2D) score plots of the cell line samples presented for the spectroscopy data: Raman (**a**) and FTIR (**b**).

**Table 1 ijms-21-01826-t001:** The Fourier Transform InfraRed (FTIR) and Raman peak positions in the analyzed cell line and TMV samples with a description of the vibrations corresponding to the respective functional groups [27,28,29,30,31,32,33,34,35,36,37,38,39,40,41,42].

**FTIR Spectroscopy Peaks (cm^−1^)**								
**HCT116**	**TMV_HCT116_**	**LoVo**	**TMV_LoVo_**	**SW480**	**TMV_SW480_**	**SW620**	**TMV_SW620_**	**Vibrations**
972	970	973	976	981	983	976	975	PO_3_^−2^ group from DNA, RNA and phospholipids [28]
1083	1083	1085	1081	1081	1080	1072	1067	C–O group from glycogen [29]
1232	1237	1239	1238	1237	1236	1234	1238	Amide III [30]
1302	1306	1307	1308	1309	1309	1309	1307	CH_2_ group from protein [31,32]
1400	1403	1402	1404	1400	1402	1402	1401	CH_2_ group from protein and lipids [33,34]
1456	1457	1458	1457	1459	1458	1458	1458	CH_2_ group from cholesterol [35]
1546	1544	1543	1547	1544	1544	1541	1542	Amide II [32]
1643	1644	1646	1646	1646	1645	1645	1643	Amide I [32]
1738	Absent	Absent	Absent	Absent	1731	1733	1741	C–O group from lipids [34]
2858	2862	2861	2861	2860	2860	2860	2858	Symmetric stretching vibrations of CH_2_ [34]
2926	2923	2927	2928	2927	2926	2924	2925	Asymmetric stretching vibrations of CH_2_ [34]
3064	Absent	3075	Absent	2963	Absent	Absent	Absent	CH_3_ asymmetric stretching [34]
3284	3290	3287	3294	3301	3284	3287	3284	Amide A and OH group from water [35]
**Raman Spectroscopy Peaks (cm^−1^)**								
**HCT116**	**TMV_HCT116_**	**LoVo**	**TMV_LoVo_**	**SW480**	**TMV_SW480_**	**SW620**	**TMV_SW620_**	**Vibrations**
1001	999	981	1000	988	987	986	984	C–H in-plane bending mode of phenylalanine [36]
1072	1080	1068	1083	1077	1080	1072	1072	Glucose triglycerides, C–C (lipid) [37,38]
Absent	Absent	Absent	Absent	1280	Absent	1281	1284	Amide III [39]
1301	1351	1291	1360	1362	1355	1362	1362	CH_3_/CH_2_ twisting or bending mode of lipid/collagen [40,41]
1450	1451	1481	1454	1479	1453	1461	1466	Fatty acids, CH_2_ (lipids and proteins) [40,42]
1668	1658	1620	1657	1647	1654	1637	1635	Amide I [43]
1784	1771	1761	1764	1786	1776	Absent	Absent	C55O ester (lipids) [42]
Absent	Absent	Absent	2700	2722	2728	Absent	Absent	Stretching vibrations of CH, NH, and OH groups [40]
2931	2933	2932	2931	2926	2929	2929	2926	CH band of lipids [42]
3145	Absent	Absent	Absent	Absent	Absent	3124	Absent	CH from cholesterol and cholesterol ester [42]

**Table 2 ijms-21-01826-t002:** Value of peak areas, percentage, and ratio of α and β secondary protein structures.

Sample	Peak Areas		Percentage [%]		α-Helix/β-Harmonica Ratio
	α-Helix	β-Harmonica	α-Helix	β-Harmonica	
**HCT116**	55.36	59.91	48.03	51.97	0.92
**TMV_HCT116_**	12.24	13.47	47.61	52.39	0.91
**LoVo**	63.34	59.40	51.61	48.39	1.07
**TMV_LoVo_**	19.70	17.80	52.53	47.47	1.11
**SW480**	33.35	35.53	48.42	51.58	0.94
**TMV_SW480_**	33.99	34.45	49.67	50.33	0.99
**SW620**	40.78	63.42	39.14	60.86	0.64
**TMV_SW620_**	22.25	19.75	52.98	47.02	1.13

**Table 3 ijms-21-01826-t003:** Discriminant analysis results using PCA-LDA and leave-one-out cross validation to distinguish between the obtained spectra.

Sensitivity	HCT116	TMV_HCT116_	LoVo	TMV_LoVo_	SW480	TMV_SW480_	SW620	TMV_SW620_
Specificity								
**HCT116**	100%	85%	94%	98%	96%	98%	95%	95%
	100%	87%	98%	94%	96%	99%	90%	64%
**TMV_HCT116_**	85%	100%	93%	92%	87%	89%	90%	96%
	87%	100%	94%	96%	95%	85%	98%	63%
**LoVo**	94%	93%	100%	89%	92%	93%	93%	98%
	98%	94%	100%	98%	99%	98%	97%	51%
**TMV_LoVo_**	98%	92%	89%	100%	96%	97%	85%	99%
	94%	96%	98%	100%	99%	95%	99%	57%
**SW480**	96%	87%	92%	96%	100%	96%	96%	98%
	96%	95%	99%	99%	100%	96%	98%	55%
**TMV_SW480_**	98%	89%	93%	97%	96%	100%	91%	99%
	99%	85%	98%	95%	96%	100%	90%	65%
**SW620**	95%	90%	93%	85%	96%	91%	100%	97%
	90%	98%	97%	99%	98%	90%	100%	68%
**TMV_SW620_**	95%	96%	98%	99%	98%	99%	97%	100%
	64%	63%	51%	57%	55%	65%	68%	100%

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
