# Peer review of "Similarities in the General Chemical Composition of Colon Cancer Cells and Their Microvesicles Investigated by Spectroscopic Methods-Potential Clinical Relevance"

_ijms, 2020, doi:10.3390/ijms21051826_

Round 1

Reviewer 1 Report

The authors examine the chemical composition of 4 colorectal cancer cell lines along with their corresponding microvesicles. They utilize two spectroscopy techniques, FTIR and Raman, and hypothesize that the similarity of the chemical composition between the TMV and the cells makes TMV useful surrogates for studying tumor cells. They determined that no observable difference exists between tumor derived microvesicles and their founding cell lines. They interpret this to mean that studying microvesicles may help identify the molecular classification of colon cancer. The paper is well written and clearly communicates their findings. Some modifications are needed to increase the readability of the paper and better describe the data.

Figure 1 and 2 characterize the TMVs from all four cell lines. Figure 2 demonstrates that the majority are 120-500 nm. Here the authors do not quantify the number of TMV isolated from cell lines. If this is known it should be included.

Figure 2 does not label the cell lines on the NTA analysis or in the figure legend.

Figure 3 Shows the FTIR spectra of the cell lines and TMVs. The text states “as expected, the spectra of TMVs were similar…” (lines 77-78). Nothing in the introduction indicates the authors were expecting this result. Please share intuition and/or mention previous findings that would cause this expectation.

Lines 85-86 refers to SW620 as shown in Figure 3C but figure 3C is labeled SW480. Please clarify.

Figure 4. The Raman spectra. Please check the accuracy of the statement in lines 103-106.

Table 1. clearly communicates the similarities and differences in the samples.

Figure 5. Clear labeling of the groups on the graph would provide for better understanding of the findings.

Figure 6. The deconvolution of the amide I FTIR region  is useful information in this process. However, figure 6 is not labeled well and the description of numbers of component bands in the text does not seem in line with what is shown in the figure. Also, more discussion of how this information is/can be used to help identify molecular subtypes.

Table 2 is again well organized and informational.

Figure 7. HCA methods are poorly described. Needs more information including, features used and interpretation of the distance between the HCT116 and TMVHCT116.

Discussion – The similarity between the cells and the TMV derived from the same cells is evident in the data presented. The authors however state that this method may offer potential clinical classification. For this to be useful the chemical composition of TMVs from different cell lines would need to be different enough from each other. This may be true but is not rigorously tested here. Additionally, TMVs from more diverse cell types need to be studied for predictive power. In order to prevent over interpretation of the data the authors should either include more cell lines, or rather, discuss this pitfall and how it might be addressed in future studies.

Reviewer 2 Report

This manuscript aims to study the global chemical composition of tumor-derived microvesicles in colon cancer cells. To answer this question, the authors tested the protein and lipid components by using ELISA, WB, chromatography as well as Raman and FTIR spectroscopy. The authors concluded that indicating that microvesicles can be used as an useful cancer biomarker. Although this study is well organized, there are a number of concerns with this manuscript.

This manuscript study HCT116, LoVo, SW480 and SW620 cell line. How about Caco-2, the human colon cancer cell line, that possesses the epithelial differentiation characteristic. How many samples are used for hierarchical cluster analysis? HCA is analyzed based on the composition of nucleic acid, phospholipids, proteins and lipids. The author can refer “Heatmap” to list compositions in different row and hierarchical cluster in different column. How many replicates in PCA analysis? The authors can consider concise Table 3. Half of the data in Table 3 are redundant. What do curve1-6 stand for in Figure 6. The explanation can be added in the legend.

Reviewer 3 Report

The manuscript entitled: "Similarities in the Global Chemical Composition of  Colon Cancer Cells and their Microvesicles Investigated by Spectroscopic Methods – Potential Clinical Relevance" addresses actual topics. This original research demonstrates the transfer of certain chemical compounds between the tumor cells via exosomes, microvesicles and highlights the importance of two analytical methods for their identification. The Raman and FTIR methods were well described, the microvesicles were characterized properly, and several data were presented in the manuscript, which support the conclusions. Using relevant biostatistics, the authors demonstrated the importance of Raman and FTIR methods in the analysis of distinct chemical patterns in microvesicles derived from colon tumor cell lines. The comparison between the results obtained through Raman and FTIR is comprehensive, providing a clear and suitable interpretation of the results.  

Comments:

Introduction chapter

- a newer reference regarding cancer incidence should be inserted instead or in addition to references 1-3

- as regards the superiority of the analytical methods vs ELISA method, they are complementary, and unlike the spectroscopy, the immune enzymatic method can identify individual proteins in the biologic samples, and not chemical groups or secondary structures.

- the comparison between the costs of Elisa testing and the modern spectroscopic methods should be better documented; the reagents are indeed more expensive for Elisa, but the equipment for spectrometry represent a considerable investment, therefore it should be employed for a very large number of testing.

Results

Figure 7- in figures 7a and b it is no information on x axis.

Some explanations and a short discussion on HCA method is needed, similar with the well-described PCA analysis.

Discussion chapter

The authors gave a logical explanations on the similarities and differences between the results obtained via FTIR and Raman, and generally, the discussion section is very logical. Still, the authors should substantiate the "Potential Clinical Relevance",  stated in the title of the manuscript. In the discussion it was made a review on several previous studies which employ molecular  methods for diagnosis and follow-up, including the CMS classification. I suggest to highlight the perspectives for the clinical use of the two methods.

Material and methods

4.9. Data analysis- the PCA and LDA analysis methods were part in OPUS 7.0 software or OriginLab? Please specify the providers of OriginLab and Past software.

Reviewer 4 Report

The manuscript Similarities in the global chemical composition of colon cancer cells and their microvesicles investigated by spectroscopic methods – potential clinical relevance by Depciuch et al. describes the applications of vibrational spectra to differentiate colon cell lines and theri corresponding TMVs.

Cell lines were selected appropriately (they represent cells with different aggressiveness and metastatic potential.
Description of Wester blot analysis is very cursory and should be expanded in the revision version of manuscript. In the part related to the chemical composition of the colon cancer cells are presented Raman and IR spectra. Are they normalized data? If thei are , whta type of normalization was used? Are they average spectra (how many spectra were used to obtain the presented results?) In the discusion of these results , summarized iin Table 1, the absents of some pvibrational peaks should be comment. also the part related to the analysis of protein structures doesn't contain the explanation why the number of components used to fitt the Amide region is different. Authors should also discussed in more detail the difference in a-helix/b-harmonica ration for SW620 and TMV-SW 620. How were obtain the data presented in PCA analysis (Figure8)? Are they avaerage spectra? (one can see only single points) The text desribing PCA results (lines 175-188) dosn't correlate well with the data presented in Table 3. On the one side Author claim that is " very difficult distinguising" some cells on the other side results of PCA_LDA show realy high parameters. In the Discussion the long part discribes IR and Raman basis and only one paper by Berg et al. is shortly discussed. In part 4.5 more information about the cells Raman and IR measurements shoud be added (for which form of cells the Raman and IR spectra were recorded ? Suspention, film?)

Round 2

Reviewer 2 Report

Thank you for revision. I agree to publish this manuscript to IJMS.